# Estimated glucose disposal rate and risk of metabolic syndrome: A population-based study

**Lin Zhang, Nian Cai, Li Mo, Xiaofang Tian, Bohai Yu***

Shenzhen Hospital (Futian) of Guangzhou University of Chinese Medicine, Shenzhen, Guangdong, China

* wwwcnwxx6688@163.com

## Abstract

### Aims

This study aims to explore the association between the estimated glucose disposal rate (eGDR) and the risk of metabolic syndrome (MetS), with a focus on the mediating role of BMI.

### Methods

Data for this study came from the 2011 and 2015 China Health and Retirement Longitudinal Study. We used multivariable logistic regression and restricted cubic splines to assess the relationship between eGDR and MetS, with subgroup and interaction analyses to identify moderating factors. The diagnostic ability of eGDR for MetS was evaluated via receiver operating characteristic curves.

### Results

In total, 3,229 participants were included, with 745 (23.07%) diagnosed with MetS. In the fully adjusted model, each interquartile range increase in eGDR was associated with a 58% reduced MetS risk (odds ratio = 0.42, 95% confidence interval: 0.36–0.49, $P < 0.001$). A significant nonlinear dose–response relationship between eGDR and MetS risk was observed ($P < 0.001$, both overall and nonlinear).Spline regression analysis revealed that the protective effect of eGDR was significant up to 11.88 mg/kg/min (standard error = 0.17). Subgroup analysis revealed significant interaction effects of marital status and residential area on the eGDR–MetS relationship ($P < 0.05$), while BMI mediated 29% of eGDR's total effect on MetS. Receiver operating characteristic analysis showed eGDR's good predictive performance for MetS, with an area under the curve of 0.71 (95% confidence interval: 0.69–0.73).

### Conclusion

Higher eGDR levels were linked to a significantly lower MetS risk, with approximately 29% of this association mediated by BMI, suggesting that individuals with low eGDR may benefit from closer monitoring for MetS development.

**Data availability statement:** The minimal dataset necessary to replicate the study findings has been deposited in the Figshare repository: 10.6084/m9.figshare.30375043. This dataset was derived from the China Health and Retirement Longitudinal Study (CHARLS). Researchers can access the original CHARLS data upon application at http://charls.pku.edu.cn/.

**Funding:** The author(s) received no specific funding for this work.

**Competing interests:** The authors have declared that no competing interests exist.

## Introduction

Metabolic syndrome (MetS), also referred to as Syndrome X or insulin resistance syndrome, represents a cluster of cardiovascular risk factors rather than a single disease [1]. The pathophysiology of MetS is intricate and not yet fully elucidated, and its diagnostic criteria have undergone continuous refinement since the 1980s [2]. Although several organizations have proposed varying diagnostic criteria, the differences between criteria are generally minor. Global estimates of MetS may vary due to differences in study design and population coverage, its prevalence is approximately three times that of diabetes, affecting roughly one-quarter of the global population, which corresponds to more than one billion individuals [3]. This syndrome not only presents significant health risks but also imposes a massive economic burden, amounting to trillions of dollars in healthcare costs and potential economic productivity losses [4]

To accurately assess the risk of MetS, researchers have developed various evaluation metrics [5,6]. Among these, the estimated glucose disposal rate (eGDR), which incorporates waist circumference (WC), hypertension, and glycated hemoglobin (HbA1c), is used to assess individual insulin sensitivity [7]. As a continuous, simple, and non-invasive measure, eGDR has been widely adopted in clinical research. Previous studies have demonstrated that lower eGDR values are associated with significantly increased risks of renal disease, peripheral vascular disease, coronary artery disease, and mortality [8–11]. However, the specific relationship between eGDR and MetS has not yet been fully established.

Therefore, this study aims to investigate the role of dynamic changes in eGDR in predicting MetS through a prospective cohort study. By systematically evaluating the impact of eGDR variations on the risk of MetS, the study seeks to provide a scientific basis for the early identification and intervention of MetS.

## Method

### Study population

The China Health and Retirement Longitudinal Study(CHARLS) is a survey targeting middle-aged and older adults in China, specifically, individuals aged 45 years and above [12]. The survey aims to establish a high-quality micro-database that encompasses a wide range of socioeconomic and health-related information to support research on aging. The national baseline survey of China Health and Retirement Longitudinal Study was conducted between 2011 and 2012, covering 28 provinces, 150 counties/districts, and 450 villages/communities. A total of 17,708 individuals from 10,257 households were interviewed. Institutionalized elderly individuals were not included in the initial sampling; however, baseline respondents who later moved into institutions were followed up in subsequent surveys. The second wave of data collection took place in 2013, with no biomarkers collected during this follow-up period. The third wave, primarily conducted between July and October 2015, included the collection of blood samples as in the first wave. The fourth wave was conducted in 2018. The Ethics Review Committee of Peking University approved the study protocol, and all participants provided written informed consent.

The study population comprised participants from the first and third waves of the survey who had available blood test data [13]. Participants were required to fast overnight before the blood draw. The blood samples were collected by nurses from local community hospitals or clinics hired by the research team. Three tubes of venous blood were collected from each participant, and after preliminary processing, the samples were transported directly to Peking University via a cold chain for further analysis.

In accordance with the study objectives, participants with incomplete baseline eGDR data (n = 7,844), a baseline diagnosis of MetS (n = 4,750), use of glucose-lowering medications (n = 71), age below 45 years or missing age information (n = 90), and incomplete MetS outcome data were excluded (n = 1,724). This left a final set of 3,229 participants for analysis (Fig 1).

## Measurement

In this study, the eGDR was calculated using a previously described formula [14]:

eGDR (mg/kg/min) = 21.158 − (0.09 × WC) − (3.407 × hypertension) − (0.551 × HbA1c), where WC was measured in cm, hypertension was coded 1 for yes and 0 for no, and HbA1c was included as a percentage [10].

## Outcome

The diagnosis of MetS was based on criteria defined by Chinese standards [15], which include five components:

(1) Central obesity: WC ≥ 80 cm for women and ≥ 90 cm for men;

(2) Elevated triglyceride (TG) levels: TG levels ≥ 150 mg/dL;

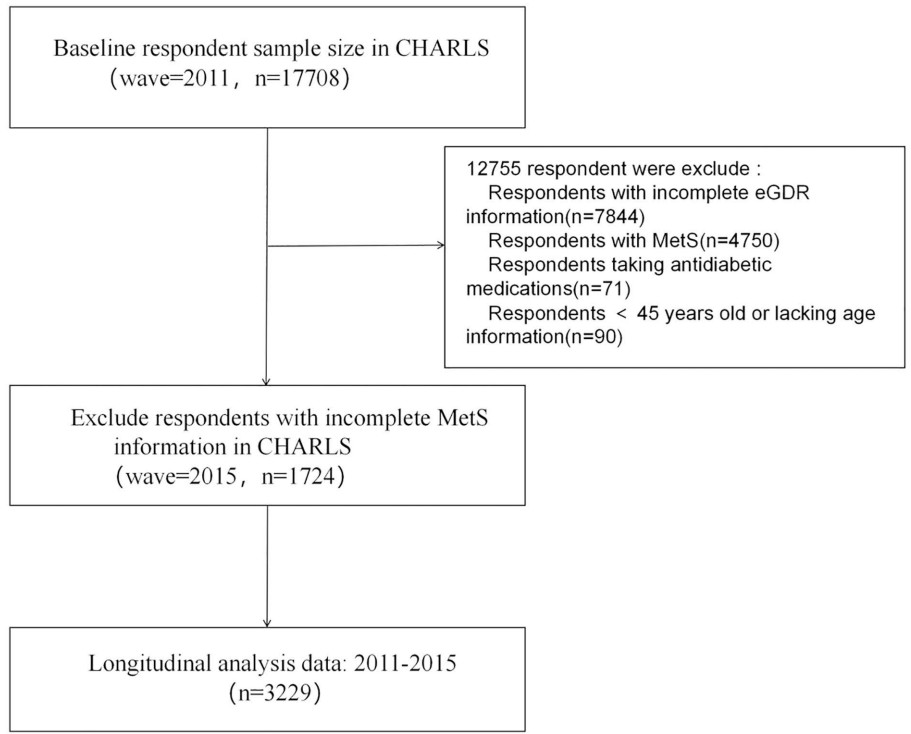

**Fig 1. Flowchart illustrating the respondents selection process.**

(3)  Low high-density lipoprotein cholesterol levels: high-density lipoprotein cholesterol levels < 40 mg/dL for men and < 50 mg/dL for women;

(4)  Elevated blood pressure (BP): systolic BP ≥ 130 mmHg and/or diastolic BP ≥ 85 mmHg, or use of antihypertensive treatment;

(5)  Elevated fasting plasma glucose (FPG): FPG levels ≥ 100 mg/dL, use of antidiabetic medication, or self-reported diabetes history.

A diagnosis of MetS is made if at least three of these five criteria are present.

## Covariate assessment

We reviewed baseline demographic information, including age, sex, residence location (village/ city/town), education level (below primary school/primary school/junior high school/high school and above), marital status (married/ non-married), and sleep duration. Basic anthropometric measurements such as height, weight, and WC were collected, along with potential risk factors including smoking status (never/former/current) and alcohol consumption status (never/former/current). Laboratory tests included FPG, TG, C-reactive protein (CRP), and HbA1c. Additionally, self-reported data on hypertension history and use of antidiabetic medications were collected.

## Statistical analyses

Baseline characteristics of non-normally distributed data were analyzed using the median and interquartile range (Q1–Q3). Categorical variables are presented as percentages. To compare data based on eGDR quartiles and the presence of MetS, we used t-tests, Mann–Whitney U tests, or chi-square tests as appropriate. Three logistic regression models were employed to calculate the odds ratios (OR) and 95% confidence intervals (CIs) for MetS, analyzing eGDR both as a continuous variable (per interquartile range increase) and as a categorical variable (quartiles). In Model 1, we assessed the crude association between eGDR and MetS. Model 2 included adjustments for age, sex, marital status, education level, residence location, sleep duration, smoking, and alcohol consumption. Model 3 included further adjustments for TG, CRP, HbA1c, and hypertension.

This study explored the dose–response relationship between eGDR and MetS, visualizing it through spline regression analysis to identify critical points and investigate the nonlinear relationship. Additionally, we conducted subgroup and interaction analyses based on age, sex, marital status, education level, residence location, smoking, alcohol consumption, and sleep duration to comprehensively assess the association between eGDR and MetS.

Additionally, we performed mediation analysis using the R package "mediation", "bda" to examine the mediating role of BMI in the relationship between eGDR and MetS. To further evaluate the diagnostic performance of eGDR in predicting MetS, we conducted receiver operating characteristic (ROC) curve analysis to determine its accuracy in diagnosing MetS.

All statistical analyses were performed using R software version 4.1, with a two-tailed $P$-value < 0.05 considered statistically significant.

## Results

### Characteristics of the study population according to MetS status

The study included a total of 3,229 participants with a mean age of 58.65 ± 8.77 years. Among them, 1,711 (53.0%) were male, and 1,518 (47.0%) were female. During the long-term follow-up from 2011 to 2015, 745 participants (23.0%) developed MetS.

Table 1 presents the baseline characteristics of the participants, categorized by the presence or absence of MetS. Participants with MetS had significantly higher levels of TG and CRP and a significantly higher BMI and WC than those

**Table 1. Baseline characteristics of study population by MetS status at follow-up.**

| | Total (n = 3229) | Non-MetS (n = 2484) | MetS (n = 745) | P |
|---|---|---|---|---|
| Age, years | 58.65 ± 8.77 | 58.53 ± 8.66 | 59.03 ± 9.11 | 0.283 |
| Sex, n (%) | | | | < 0.001 |
| Female | 1518 (47.01) | 1089 (43.84) | 429 (57.58) | |
| Male | 1711 (52.99) | 1395 (56.16) | 316 (42.42) | |
| Marital status, n (%) | | | | 0.066 |
| Non-Married | 460 (14.25) | 338 (13.61) | 122 (16.38) | |
| Married | 2769 (85.75) | 2146 (86.39) | 623 (83.62) | |
| Location, n (%) | | | | 0.114 |
| City/Town | 891 (27.59) | 668 (26.89) | 223 (29.93) | |
| Village | 2338 (72.41) | 1816 (73.11) | 522 (70.07) | |
| Education, n (%) | | | | 0.564 |
| Below primary school | 1538 (47.68) | 1167 (47.00) | 371 (49.93) | |
| Primary school | 757 (23.47) | 588 (23.68) | 169 (22.75) | |
| Middle school | 653 (20.24) | 511 (20.58) | 142 (19.11) | |
| High school and above | 278 (8.62) | 217 (8.74) | 61 (8.21) | |
| Drink status, n (%) | | | | < 0.001 |
| Never | 1777 (55.07) | 1329 (53.52) | 448 (60.22) | |
| Former | 243 (7.53) | 182 (7.33) | 61 (8.20) | |
| Current | 1207 (37.40) | 972 (39.15) | 235 (31.59) | |
| Smoke status, n (%) | | | | < 0.001 |
| Never | 1803 (55.99) | 1336 (53.89) | 467 (63.02) | |
| Former | 250 (7.76) | 197 (7.95) | 53 (7.15) | |
| Current | 1167 (36.24) | 946 (38.16) | 221 (29.82) | |
| Sleep time, hours | 6.33 ± 1.89 | 6.33 ± 1.88 | 6.33 ± 1.92 | 0.795 |
| Hypertension, n (%) | | | | < 0.001 |
| No | 2871 (88.91) | 2307 (92.87) | 564 (75.70) | |
| Yes | 358 (11.09) | 177 (7.13) | 181 (24.30) | |
| TG (mg/dL) | 92.56 ± 39.43 | 89.77 ± 38.93 | 101.88 ± 39.67 | < 0.001 |
| HbA1C (%) | 5.1 ± 0.41 | 5.09 ± 0.40 | 5.13 ± 0.43 | 0.192 |
| CRP (mg/L) | 1.54 ± 2.36 | 1.49 ± 2.28 | 1.70 ± 2.61 | < 0.001 |
| BMI (kg/m²) | 22.03 ± 3.35 | 21.60 ± 3.22 | 29.47 ± 3.38 | < 0.001 |
| WC (cm) | 80.49 ± 8.16 | 79.32 ± 7.65 | 84.39 ± 8.6 | < 0.001 |
| eGDR (mg/kg/min) | 10.81 ± 1.48 | 11.06 ± 1.31 | 10.01 ± 1.71 | < 0.001 |

Data are presented as n (%) for categorical variables and as median (interquartile range, IQR) or mean ± standard deviation (SD) for continuous variables, depending on the distribution. P-values were calculated from chi-square tests (categorical variables) or rank-sum tests (continuous variables without normal distribution), or t-tests (continuous variables with normal distribution).

TG, Triglyceride; HbA1C, Glycated Hemoglobin A1C; CRP, C-Reactive Protein; WC, Waist Circumference; eGDR, Estimated Glucose Disposal Rate; MetS, Metabolic Syndrome.

without MetS (P < 0.001). The prevalence of MetS was also significantly higher among female participants (57.58% vs. 42.42%, P < 0.001). There were statistically significant differences between the MetS and Non-MetS groups in terms of smoking (63.02% vs. 53.98%, P < 0.001), drinking (60.22% vs. 53.52%, P < 0.001), and hypertension (75.70% vs. 24.30%, P < 0.001). Notably, we observed that individuals with MetS had significantly lower eGDR levels compared with those without MetS (10.43 vs. 11.19, P < 0.001).

## Characteristics of study population by eGDR quartiles

Furthermore, we conducted an analysis of the population across different eGDR quartiles (Table A1 in S1 File). Notably, as eGDR levels increased, there was a corresponding decline in the incidence of newly diagnosed MetS, with rates of 44.68%, 23.05%, 14.00%, and 10.53% across increasing quartiles (P<0.001). Additionally, significant differences were observed among participants across various baseline characteristics, including age, gender, marital status, residence location, hypertension status, smoking, TG, CRP, HbA1c, BMI, and WC (all P<0.05).

## Associations between baseline eGDR and MetS incidence

Table 2 outlines the association between eGDR levels and the risk of developing MetS. There was a significant inverse relationship between eGDR and the risk of MetS. Specifically, in the unadjusted model, compared with the lowest quartile of eGDR (Q1), participants in the second (Q2), third (Q3), and fourth (Q4) quartiles exhibited a progressively lower risk of MetS, with ORs and 95% CIs of 0.37 (95% CI: 0.30–0.46), 0.20 (95% CI: 0.16–0.26), and 0.15 (95% CI: 0.11–0.19), respectively. In Model 2, after adjusting for different covariates, the OR for the Q3 group increased significantly to 1.01 (95% CI: 1.00–1.02), consistent with the findings in Model 3.In the fully adjusted model (Model 3), compared with the Q1 group, the Q2 and Q4 groups had a significantly lower risk of MetS (OR = 0.50 [95% CI: 0.45–0.57] and OR = 0.44 [95% CI: 0.33–0.59], respectively), with the trend analysis showing a statistically significant effect (P for trend<0.001).

Fig 2 illustrates the dose–response relationship between eGDR and the risk of developing MetS. There was a significant nonlinear association between eGDR levels and the risk of MetS (P<0.001 [overall], P<0.001 [nonlinear]).According to the spline regression analysis, there was a strong negative association between eGDR and MetS up to an eGDR of 11.88 (standard error=0.17). Beyond this point, the association weakened, with little to no additional protective effect observed (S1 Fig).

## Subgroup analysis

To explore whether the association between eGDR and the incidence of MetS varies across different subgroups, we conducted analyses stratified by participants' socioeconomic characteristics and medical history (Fig 3). There were significant interactions between eGDR and marital status and residence location (P<0.05), with unmarried individuals and rural residents deriving a greater benefit from higher eGDR levels. Specifically, each IQR increase in eGDR was associated with a 66% reduction in the risk of MetS among rural residents (OR = 0.34 [95% CI: 0.29–0.42]) compared with a 41% reduction in urban residents (OR = 0.59 [95% CI: 0.46–0.75]). Similarly, each IQR increase in eGDR was associated with a 67% reduction in the risk of MetS among unmarried individuals (OR = 0.33 [95% CI: 0.21–0.53]) compared with a 57% reduction among married individuals (OR = 0.43 [95% CI: 0.36–0.50]).

**Table 2. Associations between baseline eGDR with follow-up incident MetS.**

|  | Model 1 | P | Model 2 | P | Model 3 | P |
|---|---|---|---|---|---|---|
| eGDR per IQR | 0.53 (0.49,0.58) | <0.001 | 0.52 (0.49,0.57) | <0.001 | 0.42 (0.36,0.49) | <0.001 |
| Q1 | Ref |  | Ref |  | Ref |  |
| Q2 | 0.37 (0.30,0.46) | <0.001 | 0.60 (0.57, 0.64) | <0.001 | 0.50 (0.45,0.57) | <0.001 |
| Q3 | 0.20(0.16, 0.26) | <0.001 | 1.01 (1.00, 1.02) | 0.161 | 1.01 (1.00,1.02) | 0.030 |
| Q4 | 0.15 (0.11, 0.19) | <0.001 | 0.45 (0.34, 0.60) | <0.001 | 0.44 (0.33, 0.59) | <0.001 |
| P for trend | <0.001 |  | <0.001 |  | <0.001 |  |

Model 1 was the unadjusted model. Model 2 was adjusted for age, gender, education level, location, marital status, smoking status, drinking status, and sleep time. Model 3 was further adjusted for hypertension, TG, CRP, and HbA1C based on Model 2.

eGDR, Estimated Glucose Disposal Rate; MetS, Metabolic Syndrome; IQR, Interquartile Range.

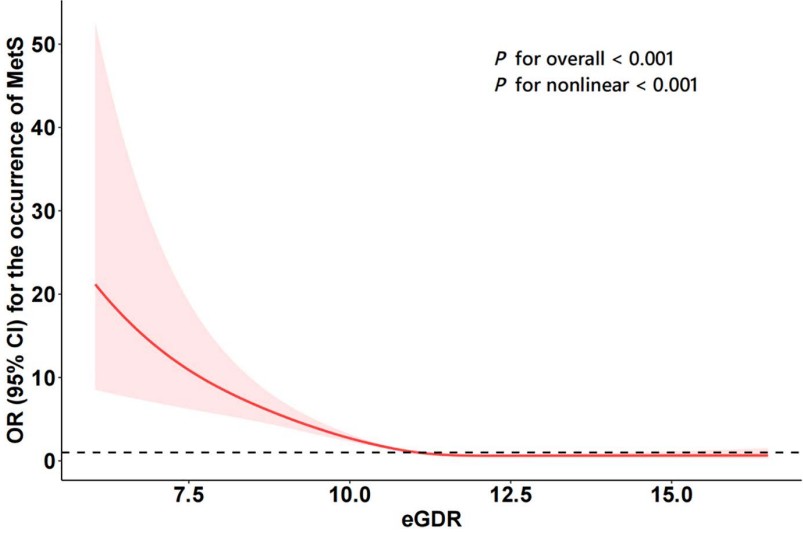

**Fig 2. Dose-response relationship between eGDR and MetS.**

## Mediation analysis

According to the mediation analysis results (Table A2 in S1 File), eGDR had a significant indirect effect on MetS through BMI (estimated average causal mediated effect = −0.00252, P < 0.001), as well as a significant and negative direct effect (estimated average direct effect = −0.00547, P < 0.001). Approximately 29.08% of the total effect of eGDR on MetS was mediated by BMI (estimated proportion mediated = 0.29077, 95% CI: 0.19421–0.42000, P < 0.001). Consistently, the Sobel test also confirmed the significance of the mediation pathway (Sobel Z = −6.63, P < 0.0001).

## ROC curve analysis

In this study, we evaluated the predictive performance of eGDR and its composite indicators (WC, HbA1c) for MetS (Table A3 in S1 File and Fig 4). The area under the ROC curve (AUC) for eGDR in predicting MetS was 0.71 (95% CI: 0.69–0.73), which was higher than that for WC (AUC = 0.67, 95% CI: 0.65–0.69) and HbA1c (AUC = 0.52, 95% CI: 0.49–0.54). These findings suggest that eGDR, as a composite indicator, has a better predictive ability than any single MetS indicator.

## Discussion

MetS is a complex cluster of metabolic disorders, and its prevalence is increasing steadily worldwide [16]. We observed a significant inverse association between eGDR and the risk of MetS. Additionally, marital status and residential location moderated this relationship, with unmarried individuals and rural residents benefiting more from higher eGDR levels. Furthermore, the dose–response analysis revealed a nonlinear negative relationship between eGDR and MetS risk. Higher eGDR levels were associated with a reduced risk of MetS up to a critical threshold of 11.88 (standard error = 0.17), beyond which the protective effect diminished. The ROC analysis further highlighted that eGDR has superior predictive power for MetS compared with single components such as WC and HbA1c.

Insulin resistance is one of the central pathological mechanisms underlying MetS [17]. Reduced insulin sensitivity leads to a diminished response to insulin, resulting in poor glycemic control. Prolonged insulin resistance not only disrupts glucose metabolism but also contributes to the development of abdominal obesity, hypertension, and dyslipidemia, key components that collectively define the criteria for diagnosing MetS [18]. Therefore, decreased insulin sensitivity (insulin resistance) significantly increases the risk of MetS. Studies show that hypertension, abdominal obesity, and elevated

| | OR (95% CI) | P for interaction |
|---|---|---|
| **Age** | | **0.081** |
| <60 | 0.47 (0.40 − 0.57) | |
| ≥60 | 0.34 (0.26 − 0.44) | |
| **Sex** | | **0.839** |
| Male | 0.43 (0.35 − 0.52) | |
| Female | 0.38 (0.30 − 0.49) | |
| **Marital** | | **0.045** |
| Non−married | 0.33 (0.21 − 0.53) | |
| Married | 0.43 (0.36 − 0.50) | |
| **Education** | | **0.542** |
| Below primary school | 0.42 (0.34 − 0.53) | |
| Primary school | 0.46 (0.34 − 0.63) | |
| Middle school | 0.44 (0.31 − 0.61) | |
| High school and above | 0.18 (0.10 − 0.34) | |
| **Location** | | **0.001** |
| Village | 0.34 (0.29 − 0.42) | |
| City/Town | 0.59 (0.46 − 0.75) | |
| **Smoking** | | **0.612** |
| Never smoker | 0.45 (0.37 − 0.54) | |
| Former smoker | 0.20 (0.09 − 0.46) | |
| Current smoker | 0.41 (0.31 − 0.53) | |
| **Drinking** | | **0.363** |
| Never drinker | 0.44 (0.36 − 0.53) | |
| Former drinker | 0.22 (0.10 − 0.46) | |
| Current drinker | 0.40 (0.31 − 0.52) | |
| **Sleep time** | | **0.161** |
| <7 | 0.42 (0.34 − 0.52) | |
| ≥7 | 0.41 (0.33 − 0.51) | |

**Fig 3. The association between eGDR and MetS varies by subgroup.**

HbA1c levels are associated with reduced insulin sensitivity, which in turn increases the risk of developing MetS [19–21]. eGDR serves as a comprehensive marker for assessing insulin sensitivity and metabolic health by integrating WC, BP, and HbA1c levels. Our study identified eGDR as a factor that protected against MetS, consistent with previous findings. A five-year follow-up study conducted by Peng Juan et al. in 956 patients with type 2 diabetes found that an eGDR > 6.34

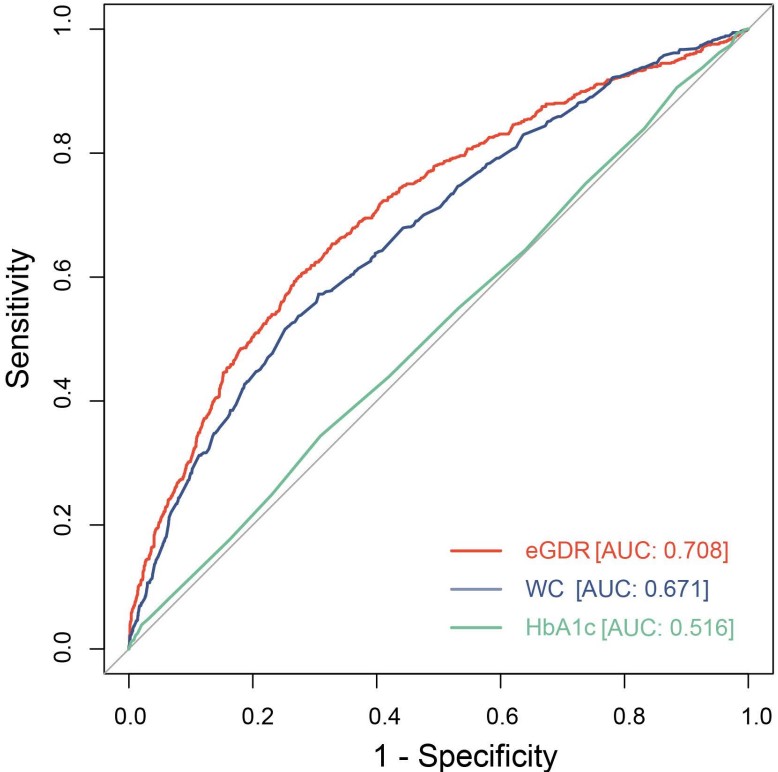

**Fig 4. Predictive analysis curves of eGDR, WC, and HbA1C for MetS.**

mg/kg/min served as a protective factor for renal outcomes [22]. Another study involving 104,697 patients with type 2 diabetes demonstrated that a higher eGDR (indicating lower insulin resistance) was associated with a lower risk of stroke and mortality [23]. Furthermore, a six-year follow-up study indicated that a higher eGDR was linked to a reduced risk of cardiovascular disease, stroke, and cardiac events [24]. These findings reinforce our conclusion that eGDR levels are significantly associated with the risk of developing MetS

Subgroup analyses indicated that married individuals and urban residents had a higher risk of MetS. For married individuals, changes in lifestyle, dietary habits, and increased social support following marriage can lead to greater energy and fat intake, which may increase the risk of obesity [25–27]. Additionally, the combined effects of aging and the added responsibilities of marriage can heighten the risk of hypertension, making married individuals more susceptible to MetS than their single counterparts [28,29]. Urban residents are also at greater risk due to factors such as prolonged sedentary work, limited opportunities for physical activity, and higher consumption of processed foods, particularly those high in sodium and low in potassium [30–32]. These factors are closely associated with an increased risk of MetS.

In this study, BMI was found to partially mediate the association between eGDR and MetS, accounting for approximately 29.08% of the total effect, suggesting that BMI serves as a bridge in this relationship. On one hand, BMI reflects overall adiposity, and higher BMI is often accompanied by reduced insulin sensitivity, negatively influencing eGDR [33,34]. On the other hand, BMI is closely linked to MetS, obesity can lead to adipose tissue dysfunction, prompting adipocytes to secrete inflammatory cytokines and adipokines, which may disrupt insulin signaling pathways and impair glucose and lipid metabolism, thereby increasing the risk of MetS [35]. Thus, BMI is not only closely related to eGDR but also partially mediates its effect on MetS through metabolic and inflammatory pathways. Moreover, the partial mediation indicates that

eGDR retains an independent direct effect on MetS, highlighting the multifactorial nature of metabolic syndrome development. Clinically, these findings underscore the importance of BMI management, weight reduction and lifestyle modification may mitigate the risk of MetS among individuals with low eGDR, and metabolic assessments should consider both insulin sensitivity and body weight control strategies.

### Strengths and limitations

Our study has several strengths. First, it is the first prospective investigation of the relationship between eGDR and MetS risk. The findings further establish eGDR as a protective factor against MetS. Second, this study is based on a large, representative national sample, which enhances the generalizability and reliability of the results. Finally, the prospective cohort study design allows some elucidation of the causal relationship between eGDR and MetS, which provides valuable insights for reducing the incidence of MetS.

Nevertheless, the study has several limitations. First, it encompasses only a limited set of variables, potentially omitting other relevant factors. Second, despite adjusting for various confounders, the analysis could not account for genetic factors, dietary patterns, and other lifestyle influences due to data constraints. Third, the study was conducted solely among a Chinese population, which limits the generalizability of the findings to other countries and regions. Future research should involve additional prospective cohort studies to further elucidate the relationship between eGDR and MetS, thereby providing more comprehensive scientific evidence for MetS prevention.

### Conclusion

In summary, our findings suggest that eGDR is significantly negatively associated with the risk of developing MetS, with BMI partially mediating this association. Improving insulin sensitivity may serve as effective strategies for the prevention of MetS. This research provides valuable insights for public health initiatives aimed at mitigating the risk of MetS.

### Supporting information

**S1 File. Supplemental tables regarding study population characteristics, mediation analysis, and predictive values.**
(DOCX)

**S1 Fig. Spline Regression, with a breakpoint for eGDR at 11.88 (SEM = 0.17).**
(TIF)

### Acknowledgments

We would like to express our sincere gratitude to Peking University for providing the CHARLS data and to all individuals involved in data collection and management.

### Author contributions

**Conceptualization:** Lin Zhang.

**Data curation:** Lin Zhang, Li Mo.

**Formal analysis:** Nian Cai, Xiaofang Tian.

**Investigation:** Nian Cai.

**Methodology:** Lin Zhang.

**Project administration:** Bohai Yu.

**Resources:** Li Mo.

**Software:** Xiaofang Tian.

**Supervision:** Nian Cai.

**Validation:** Li Mo.

**Visualization:** Xiaofang Tian.

**Writing – original draft:** Lin Zhang.

**Writing – review & editing:** Bohai Yu.

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
