## [Decision Letter · Decision Letter 0]

23 Aug 2025

Dear Dr.  Yu,

Thank you for submitting your manuscript to PLOS ONE. After careful consideration, we feel that it has merit but does not fully meet PLOS ONE’s publication criteria as it currently stands. Therefore, we invite you to submit a revised version of the manuscript that addresses the points raised during the review process.

We look forward to receiving your revised manuscript.

Kind regards,

Marwan Salih Al-Nimer, MD, PhD

Academic Editor

PLOS ONE

Journal Requirements:

4. Thank you for uploading your study's underlying data set. Unfortunately, the repository you have noted in your Data Availability statement does not qualify as an acceptable data repository according to PLOS's standards.

5. Please amend your manuscript to include your abstract after the title page.

Additional Editor Comments:

**major revision**

The authors

a few points need rephrasing and clarifications

1: The objectives of the study is focusing on the role of BMI as a mediator for the development of MetS. The conclusion in the abstract and at the end of discussion is free from this assumption

2: Typing errors, e.g., recheck Table 1

3: Explain the units of the eGDR formula, i.e., the unit kg is derived from....., min is derived from......

4: The results in the tables missed the "the data presented as median (IQR)".

5: The title of figures was typed as capitalized each word

6: The authors focused on the mediation (indirect) effect), why they did not use Sobel test, which is more accurate.

7: The typing of the references in the text and in the references section need to recheck.

Reviewer's Responses to Questions

**Comments to the Author**

1. Is the manuscript technically sound, and do the data support the conclusions?

Reviewer #1: Yes

Reviewer #2: Yes

2. Has the statistical analysis been performed appropriately and rigorously?

Reviewer #1: Yes

Reviewer #2: Yes

3. Have the authors made all data underlying the findings in their manuscript fully available?

Reviewer #1: Yes

Reviewer #2: No

4. Is the manuscript presented in an intelligible fashion and written in standard English?

Reviewer #1: Yes

Reviewer #2: Yes

Reviewer #1: Thank you for the opportunity to review this manuscript. I have provided my detailed answers to the questions above in the space below. I found the study to be of high relevance and scientific value, and I have no concerns regarding dual publication, research ethics, or publication ethics at this time.

Reviewer #2: To improve transparency and reproducibility, the authors should be encouraged to provide a clear data availability statement, detailing how and where the dataset and code used for analysis (e.g., R scripts) can be accessed or requested.

**Do you want your identity to be public for this peer review?** For information about this choice, including consent withdrawal, please see our Privacy Policy

Reviewer #1: No

Reviewer #2: **Yes: ** Melchor Alpízar Salazar

---

## [Author Response · Author response to Decision Letter 1]

10 Oct 2025

Dear Editor and Reviewers,

We would like to express our sincere gratitude for your thoughtful comments and constructive suggestions on our manuscript entitled Estimated Glucose Disposal Rate and Risk of Metabolic Syndrome: A Population-Based Study. Your insightful feedback has been invaluable in guiding the improvement of our work. We deeply appreciate the time and effort you have invested in reviewing our manuscript. In response to your comments, we have made revisions to address each point raised. The changes made to the original manuscript are highlighted in red, and a detailed point-by-point response follows below.

Reviewer 1

Q1: The title is informative but could be refined slightly. Consider simplifying it for better clarity and flow, e.g.:

"Higher Estimated Glucose Disposal Rate Is Associated with Reduced Risk of Metabolic Syndrome: A Population-Based Study"

A1: Thank you very much for your valuable suggestion regarding the title. We have ultimately decided to adopt the following title: Estimated Glucose Disposal Rate and Risk of Metabolic Syndrome: A Population-Based Study. This title clearly conveys the core relationship between the exposure and the outcome, while also meeting the journal’s requirements for clarity and conciseness.

Q2: The abstract is clear and well-structured, but it slightly exceeds standard journal lengths. Consider shortening by focusing only on the most critical results and interpretations.

Line 30: “Spline regression analysis further revealed...” Consider removing "further" for conciseness unless referring back to a prior method.

Line 37: "suggesting individuals with low eGDR should monitor their potential..." This phrasing is slightly vague. Consider rewording to: "suggesting that individuals with low eGDR may benefit from closer monitoring for MetS development."

A2�Thank you very much for your careful review of my abstract and for providing such valuable suggestions. PLOS ONE requires that abstracts not exceed 300 words, and I have now reduced mine to 263 words, making it more concise. I have also revised Line 32 and Lines40-41 as suggested. I sincerely appreciate your hard work and thoughtful guidance on my manuscript.

Q3�Line 48–49: The phrase “Despite the limited availability of global data on MetS...” might be misleading since MetS prevalence is relatively well-documented globally.

A3�Thank you very much for your detailed suggestions. I realized that my original wording was inaccurate. After carefully reviewing the literature, I have revised the phrasing. It is now stated as: “Global estimates of MetS may vary due to differences in study design and population coverage, its prevalence is approximately three times that of diabetes, affecting roughly one-quarter of the global population, which corresponds to more than one billion individuals.” Thank you for your understanding.(Lines50-56)

Q4�Line 100: In the formula for eGDR, clarify the hypertension coding. Currently, it says "1 for yes and 2 for no," which might be an error. Standard binary coding is usually 1 = yes, 0 = no.

A4�I apologize for my oversight, and thank you for your reminder. I have corrected the error and included the corresponding reference.(Line105)

Q5: Line 151: “The study included a total of 3,229 participants with a mean age of 58 years.”Consider adding SD if available to clarify variation.

A5: We thank the reviewer for the valuable suggestion regarding the presentation of age. In the revised manuscript, we have updated the text to report the total participants’ age as mean ± standard deviation: “The study included a total of 3,229 participants with a mean age of 58.7 ± 8.8 years.”

We kept the age data in Table 1 as median (interquartile range), since the distribution of age is slightly skewed, and median (IQR) is more appropriate for summarizing the groups. We believe this approach addresses the reviewer’s comment while accurately reflecting the data.

We appreciate the reviewer’s careful reading and helpful comment.(Lines156, table1, Lines170-171)

Q6: Line 186: “the OR for the Q3 group was no longer significantly lower..." This needs clarification. The OR seems to be near 1.01 with a CI of [1.00–1.02], yet still has a significant P-value. Consider double-checking if this is an error or needs better explanation.

A6: Thank you for your careful observation and feedback. I apologize for my oversight and have revised the sentence accordingly.”In Model 2, after adjusting for different covariates, the OR for the Q3 group increased significantly to 1.01 (95% CI: 1.00–1.02), consistent with the findings in Model 3.”(Lines192-196)

Reviewer 2

Q1: The objectives of the study is focusing on the role of BMI as a mediator for the development of MetS. The conclusion in the abstract and at the end of discussion is free from this assumption

A1�We sincerely thank the reviewer for this insightful comment. In response, we have revised the conclusion of the abstract, the discussion section, and the final conclusion of the manuscript to explicitly highlight the mediating role of BMI in the relationship between eGDR and MetS. We greatly appreciate the reviewer’s careful consideration and constructive feedback, which have made our manuscript more professional and rigorous. Thank you again for your valuable efforts in improving our work.(Lines40-41, Lines289-303, Line322)

Q2: Typing errors, e.g., recheck Table 1

A2�Thank you for your careful observation. I apologize for my oversight. I have carefully reviewed and revised Table 1 to ensure it is more accurate, standardized, and complete.

Q3: Explain the units of the eGDR formula, i.e., the unit kg is derived from....., min is derived from......

A3�Thank you for this insightful comment. The eGDR formula was originally derived from regression models against the euglycemic–hyperinsulinemic clamp, which serves as the gold standard for insulin sensitivity measurement (expressed in mg/kg/min). In this equation, waist circumference is entered in centimeters, hypertension is coded as 1 for yes and 0 for no, and HbA1c is entered as a percentage. Because the regression coefficients were estimated against clamp data expressed in mg/kg/min, the final calculated eGDR values naturally inherit this unit. We have clarified this point in the revised manuscript and greatly appreciate the reviewer’s constructive suggestion.(Line106)

Q4: The results in the tables missed the "the data presented as median (IQR)".

A4�We sincerely thank the reviewer for the valuable suggestion regarding the presentation of continuous variables in Table 1. In the revised manuscript, we have updated the table footnote to clearly indicate that continuous variables are presented as median (interquartile range, IQR) or mean ± standard deviation (SD), depending on the distribution. We believe this revision improves the clarity and accuracy of the data presentation. We greatly appreciate the reviewer’s careful reading and helpful comment.(Lines170-171)

Q5: The title of figures was typed as capitalized each word

A5: Thank you very much for your careful review and valuable comments. I apologize for the oversight regarding the figure titles. Following your suggestion, I have revised all figure titles to sentence case and carefully rechecked the overall formatting of the manuscript to avoid similar issues. I sincerely appreciate your time and effort in reviewing my work.

Q6: The authors focused on the mediation (indirect) effect), why they did not use Sobel test, which is more accurate.

A6�Thank you very much for your professional suggestion. In our study, we primarily adopted the bootstrap method implemented in the mediation package, as it is more robust in finite samples and provides estimates of the indirect effect size as well as the proportion mediated.

To address the reviewer’s comment, we additionally conducted the Sobel test. The results also demonstrated a significant mediation effect (Z = −6.63, P < 0.0001), which is consistent with the bootstrap findings.

Both methods reached the same conclusion, supporting that BMI plays a significant partial mediating role in the relationship between eGDR and MetS. The corresponding sentence has been added in the revised manuscript at Lines 235–236: “Consistently, the Sobel test also confirmed the significance of the mediation pathway (Sobel Z = −6.63, P < 0.0001).”

Once again, we sincerely thank the reviewer for this valuable suggestion, which has made our manuscript more rigorous.

(Lines235-236)

Q7: The typing of the references in the text and in the references section need to recheck.

A7�Thank you very much for your meticulous review and for pointing out the issues regarding the references. I sincerely apologize for the oversight in both the in-text citations and the reference list. Following your valuable suggestion, I have carefully rechecked every citation in the text and ensured that each one exactly matches the corresponding entry in the reference list. In addition, I have thoroughly revised the reference section to strictly adhere to the journal’s formatting requirements, including citation style, punctuation, capitalization, and consistency. As required, PLOS uses the “Vancouver” style, as outlined in the ICMJE sample references, and I have ensured that all references now fully comply with this standard. I truly appreciate your thoughtful attention to such important details, which has significantly improved the overall quality and readability of the manuscript. Thank you again for your time, effort, and dedication to reviewing my work.

---

## [Editor Report · Decision Letter 1]

13 Oct 2025

Estimated Glucose Disposal Rate and Risk of Metabolic Syndrome: A Population-Based Study

PONE-D-25-26075R1

Dear Dr. Bohai Yu,

We’re pleased to inform you that your manuscript has been judged scientifically suitable for publication and will be formally accepted for publication once it meets all outstanding technical requirements.

Kind regards,

Marwan Salih Al-Nimer, MD, PhD

Academic Editor

PLOS ONE

Additional Editor Comments (optional):

No comments
---

## [Editor Report · Acceptance letter]

PONE-D-25-26075R1

PLOS ONE

Dear Dr. Yu,

I'm pleased to inform you that your manuscript has been deemed suitable for publication in PLOS ONE. Congratulations! Your manuscript is now being handed over to our production team.

Kind regards,

on behalf of

Professor Marwan Salih Al-Nimer

Academic Editor

PLOS ONE